# Identification and Characterization of Differentially Expressed IgM Transcripts of Channel Catfish Vaccinated with Antigens of Virulent *Aeromonas hydrophila*

**Dunhua Zhang *, Miles D. Lange, Craig A. Shoemaker and Benjamin H. Beck**

Aquatic Animal Health Research Unit, Agricultural Research Service, USDA, 990 Wire Road, Auburn, AL 36832, USA; miles.lange@usda.gov (M.D.L.); craig.shoemaker@usda.gov (C.A.S.); Benjamin.beck@usda.gov (B.H.B.)
* Correspondence: dunhua.zhang@usda.gov

**Abstract:** Channel catfish (*Ictalurus punctatus*) is the top species produced in US aquaculture and motile *Aeromonas* septicemia, caused by virulent *Aeromonas hydrophila* (vAh), is one of the most severe diseases that afflict catfish farms. Previously, vaccination of fish with extracellular proteins (ECP) of vAh was shown to produce a robust antibody-mediated immune response against vAh infection. In this study, we analyzed IgM transcripts that were differentially expressed in the head kidney and liver of ECP-immunized and mock-immunized (control) fish with emphasis on a variable domain of heavy chain. Quantitative PCR analysis indicated that immunized fish produced significantly more IgM transcripts than control fish. Full-length IgM heavy chain cDNA was cloned, which encoded typical IgM peptide, including signal peptide, variable domain (VH), constant domain (CH), and carboxyl terminal peptide. Great sequence diversity was revealed in a VH segment, with the third complementarity diversity region (CDR3) being most variable. Using germline VH gene grouping method, variants (clones) of VH characterized in this study belonged to nine VH families. The most unique variants (approximately 49%) were found in the VH2 family. Vaccinated fish apparently had more unique variants than in the control fish. There were 62% and 79% of unique variants in the head kidney and liver of vaccinated fish, respectively, while 44% and 27% unique variants in the head kidney and liver of control fish, respectively. Among the unique variants in VH2 family, approximately 87% of them were found in vaccinated fish. Two-dimensional gel electrophoresis of semi-purified IgM protein confirmed that matured IgM protein was as variable as IgM transcripts identified in this study, with isoelectric points crossing from 6 to 10. Results of this study provided insight into the molecular and genetic basis of antibody diversity and enriched our knowledge of the complex interplay between antigens and antibodies in Ictalurid catfish.

**Keywords:** *Aeromonas hydrophila*; immunization; catfish IgM; variable domain; qPCR; 2D electrophoresis

## 1. Introduction

Immunoglobulins (Ig) in teleosts are highly specialized glycoproteins (antibodies) that are developed to recognize various antigens, especially those from harmful microbes, and recruit other cells and molecules to destroy invading pathogens, playing an important role in adaptive immunity [1–3]. Three isotypes of Ig, IgM, IgD, and IgT/Z, have been identified from the genomic sequences of different teleost fish species [4,5]. In channel catfish, IgM and IgD have been found in the genome and IgM appears to be the major isotype that responds to pathogens [6–8].

IgM consists of equimolar amounts of heavy ($\mu$/H) and light (L) chains arranged in a basic unit containing 2 $\mu$/H chains and 2 L chains and in a tetrameric homolog form, $(\mu/H_2L_2)_4$, in teleosts [1,9,10]. The ability to bind to a great diversity of antigens was attributed to the diversity of the variable domain (VH) of IgM. Sequencing of the catfish germline genome revealed that there is approximately a 1 Mb locus that encodes IgM

variable and constant domains: while there are 4 exons (*Cµ1-Cµ4* or *CH1-CH4*) encoding a constant domain (CH) and 2 exons for transmembrane sequences in the carboxyl terminus of IgM, there are gene segments that encode 55 variable (*Vh*), 6 diversity (*Dh*), and 12 joint (*Jh*) gene segments, the combination of which creates a functional VH in the amino terminus [6,11]. Upon exposure to an antigen, germline-coded individual *V*, *D*, and *J* genes are selected from the repertoire; they undergo alternation within B cells via somatic recombination, a process that deletes some randomly distributed number of nucleotides on their boundaries then joins them together with random non-templated nucleotides to form a specific IgM transcript [10,12,13]. Based on the relative similarity of cDNA sequences, catfish IgM heavy chain VHs were grouped into 13 families (VH1-VH13) with each family having 2 to 32 members [6,10]. However, knowledge of the expression of specific IgM genes in response to specific immunogens is limited and those antigen-driven IgM genes that are expressed from the catfish repertoire upon exposure to vAh antigens is of interest.

Previously, we observed that serum from channel catfish vaccinated with extracellular proteins (ECP) prepared from virulent *Aeromonas hydrophila* (vAh) recognized specific vAh proteins and aggregated vAh cells, resulting in immune protection of fish against the establishment of pathogenesis [14]. IgM proteins that elicited agglutination of both ECP and cells of vAh were isolated from the antiserum, revealing that there were versatile IgM proteins targeting numerous antigens (including virulence associated proteins) [15]. The aim of this study was to analyze IgM transcripts (cDNA) that were differentially expressed in tissues (head kidney and liver) between ECP-immunized and mock-immunized catfish with a particular emphasis on the variable domains of IgM heavy chain.

## 2. Materials and Methods

Fish tissues and serum protein used in this study were collected and preserved from previous experiments [15]. Briefly, experimental fish (average weight of 35 g) were immunized with ECP or mock-immunized with phosphate-buffered saline (PBS); 3 weeks after immunization, all fish were challenged with fatal dose (approximately $2 \times 10^7$ colony forming units per fish) of virulent *A. hydrophila* (vAh). All ECP-immunized fish survived the challenge while all mocked-immunized fish died. Fish in 2 replicate groups (ECP-immunized and mock-immunized; $2 \times 12$ fish each) were subjected to tissue and blood collection. The use of fish in this study was approved by the Institutional Animal Care and Use Committee of the USDA-ARS Aquatic Animal Health Research Unit.

### 2.1. RNA Isolation, cDNA Synthesis and Primers

Total RNA of head kidney and liver tissues, dissected from ECP-immunized and mock-immunized fish (each set from 24 fish was pooled) was separately extracted using the RNeasy Midi Kit (Qiagen, Germantown, MD, USA). Purified RNA was quantified with NanoDrop ND-1000 (Wilmington, DE, USA) and stored at $-80\,°C$ until use. A total of 2 sets of cDNA were prepared for each RNA sample: (1) cDNA for quantitative real-time PCR (qPCR) was synthesized using PrimeScript RT Reagent Kit (Takara, San Jose, CA, USA) and (2) cDNA ($5'/3'$-RACE-ready) for IgM transcripts was synthesized using RACE reagent (SMARTer® RACE $5'/3'$ Kit, Takara USA). The primers used in this study were listed in Table 1, in which all primers for IgM analysis were designed within the consensus sequence (within CH1 and CH2) of conserved heavy chain constant domain (Accession#M27230/A45804/AAA79003; [6,16]). All cDNA products were stored at $-20\,°C$ until use.

**Table 1.** Primers used in this study [1].

| Primer | Gene/Function | Sequence (5′ to 3′) |
|--------|---------------|---------------------|
| Ch1f-IF | IgM/3′-RACE | gattacgccaagcttCGTCGTGCAATACCCGGCGGTGCAAGC |
| Ch1r-IF | IgM/5′-RACE | gattacgccaagcttGCAGACGAAGGTAGCTGTTCCATTGTC |
| Ch2f | IgM/qPCR | ATCCCTGTTTCCCGTGTGGCA |
| Ch2r | IgM/qPCR | TCCCGCTCGCATCCTTCCATA |
| Ch2r-IF | IgM/V-domain | gattacgccaagcttCTCCCGCTCGCATCCTTCCATATG |
| Ch3r-IF | IgM/full | gattacgccaagcttTCAAGCATTAATAGCAGACAACACA |
| 18Sf | 18S rRNA/qPCR | TTGATAACCTCGGGCCGATCG |
| 18Sr | 18S rRNA/qPCR | CGTTACCCGTGGTCACCATG |

[1] The part of sequence in small letters indicates the adaptor for in-fusion cloning.

## 2.2. Cloning of Full-Length IgM Transcripts

PCR amplification of the heavy chain cDNA was performed using primer pairs, Ch1r and UPM (RACE), for the 5′ segment and primer pairs, Ch1f and UPM, for the 3′ segment. The sequence, gattacgccaagctt, added to the 5′ ends of Ch1r-IF and Ch1f-IF was for in-fusion cloning with pRACE vector (RACE). RACE-ready cDNA from head kidney tissue of mock-immunized catfish were used as templates. Touchdown PCR program, recommended by the RACE manual, was adopted: 5 cycles of 94 °C for 30 s and 72 °C for 3 min; 5 cycles of 94 °C for 30 s, 70 °C for 30 s, and 72 °C for 3 m; and 25 cycles of 94 °C for 30 s, 68 °C for 30 s, and 72 °C for 3 m. Amplicons were separated with 0.8% agarose gel. Target bands were excised from the gel and purified using Gel Extraction Kit (Qiagen). Purified amplicons were ligated to linearized pRACE vector in accordance with the protocol of RACE (Takara). Recombinant vector plasmids in ligation reaction were transformed to Stellar competent cells (RACE) and transformants were cultured on LB (Luria-Bertani) agar containing ampicillin (100 μg/mL). Plasmids propagated in individual transformed cell colonies were purified for sequence analysis. Based on the 3′ end sequence generated from 3′ RACE, another reverse primer, Ch3r-IF which is located in the 3′ untranslated region, was paired with UPM to amplify and clone the full-length IgM transcript from the 5′ RACE-ready cDNA using the methods described above.

## 2.3. Quantitative Real-Time PCR

Gene transcripts of IgM were analyzed by qPCR with cDNA templates prepared from head kidney and liver tissues in ECP-immunized and mock-immunized catfish (described above). Sequences of primer pairs, Ch2f and Ch2r, were shown in Table 1. Channel catfish 18S rRNA gene was used as an internal calibrator for normalization. PCR was performed in triplicate for all samples in a volume of 20 μL mixture, containing 2 μL primer pair (5 μM each), 8 μL diluted cDNA (equivalent to approximately 15 ng RNA) and 10 μL 2 × iQ-SYBR (Bio-RAD, Hercules, CA, USA). Amplification was conducted using a Chromo 4 real-time PCR detection system (Bio-RAD), with the following cycling parameters: one cycle of 95 °C for 3 min and 40 cycles of 95 °C for 15 s and 60 °C for 30 s. The threshold cycle ($C_T$) values of target genes were compared to the $C_T$ of 18S RNA gene.

## 2.4. Cloning of the Variable Domain of IgM

With 5′-RACE-ready cDNA as template, primer pairs, Ch2r-IF (within IgM $C\mu1$) and UPS (universal primer short, RACE), were used to amplify the variable domain of IgM. PCR reactions were conducted in a volume of 50 μL containing 4 μL cDNA, 2 μL primer pairs (10 μM each), 25 μL 2× SeqAmp PCR buffer, and 1 μL SeqAmp DNA polymerase (Takara). PCR cycling conditions were set as follows: 94 °C for 1 min and then 30 cycles of 98 °C for 10 s, 60 °C for 15 s, and 68 °C for 45 s. Amplicons were gel purified and ligated to pRACE vector to generate recombinant plasmids for sequencing analysis as described above.

### 2.5. Two-Dimensional Gel Electrophoresis of IgM Protein

To test the isoelectric point (pI) of mature IgM protein, two-dimensional gel electrophoresis was performed. The IgM protein was isolated/semi-purified from ECP-immunized fish serum as described in previous study [15]. The methods of ZOOM IPGRunner System (Invitrogen, Carlsbad, CA, USA) were used in accordance with the manufacturer's recommendations. Briefly, 30 μL IgM protein (approximately 55 μg) were mixed with 110 μL Sample Rehydration Buffer (containing 12 g urea, 0.5 g CHAPS, 63 μL pH 6–9 and 63 μL pH 9–11 ampholytes, 0.5 mL of 0.1% bromophenol blue solution and 20 mM DTT in a total volume of 20 mL). Immobilized pH gradient (IPG) strips of pH 6–10 were used to perform isoelectric focusing at voltages 200 V for 20 m, 450 V for 15 m, 750 V for 15 m, and 2000 V for 30 m. The second dimensional electrophoresis was conducted using NuPAGE 4–12% Bis-Tris ZOOM Protein Gel and MES-SDS running buffer (Novex). Protein bands in gel were revealed by staining with SimplyBlue™ solution (Invitrogen).

### 2.6. Data Processing and Analysis

Sequence alignment and analysis (such as calculation of molecular weight and isoelectric point) were performed using Vector NTI (Invitrogen,) and/or SnapGene V5.3 (www.snapgene.com). The online web service, https://blast.ncbi.nlm.nih.gov/Blast.cgi, was used to search and compare reference sequences, and http://www.cbs.dtu.dk/services/SignalP was used to predict signal peptide. Phylogenetic analysis of IgM VH sequences was conducted using neighbor joining method with 1000 replicate bootstrap test in MEGA6 (www.megasoftware.net). The relative expression levels of target genes in qPCR were determined by subtracting the mean $C_T$ of the calibrator gene by that of the target gene ($\Delta C_T = C_T$ target gene $- C_T$ calibrator). The relative changes of gene expression levels between samples of immunized (Imm) and mock-immunized (mock) tissues were calculated by $2^{-\Delta\Delta CT}$, where $\Delta\Delta C_T = \Delta C_T$ Imm $- \Delta C_T$ mock [17]. Differences of gene expression levels were analyzed with paired t test using Graph Prism 6 software. Probabilities of 0.05 or less were considered statistically different.

## 3. Results

### 3.1. Expression of IgM Transcripts

All individual qPCR samples showed a single peak following melt curve analysis and amplicons had a prospective size (Figure S1). Significantly more IgM transcripts were observed in ECP-immunized head kidney and liver tissues than those in mock-immunized tissues (Table 2). Based on comparative quantification ($2^{-\Delta\Delta CT}$), the fold changes were 1.60 (or a 60% increase) and 1.46 (or a 46% increase) for immunized head kidney and liver, respectively.

**Table 2.** Expression of IgM in kidney and liver tissues between ECP-immunized and mock-immunized channel catfish.

| Tissues | Immunization | Average $C_T \pm$ SD | Relative to Calibrator ($\Delta C_T$) | Fold Change ($2^{-\Delta\Delta CT}$) |
|---|---|---|---|---|
| Head Kidney | ECP | $18.78 \pm 0.06$ | 9.13 | 1.60 * |
| | Mock | $18.52 \pm 0.03$ | 9.81 | - |
| Liver | ECP | $20.12 \pm 0.13$ | 10.11 | 1.47 * |
| | Mock | $20.67 \pm 0.09$ | 10.67 | - |

Calibrator: 18S rRNA; $\Delta C_T = C_T$ target gene $- C_T$ calibrator; $\Delta\Delta C_T = \Delta C_T$ (ECP) $- \Delta C_T$ (Mock). Significant difference ($p < 0.05$) was marked by an asterisk.

### 3.2. The Full-Length IgM cDNA

All sequences obtained from 3′ RACE were fully aligned and matched with the reference sequence (A45804) and some partial sequences deposited in the NCBI database except for one sequence that had one base substitution from CCC (coding proline) to CAC

(coding histidine). Translated amino acid peptides were shown to encode an IgM heavy chain constant domain. Sequences from 5′ RACE were identified as the IgM heavy chain variable domain but most shared less than 70% identity of each other. A representative full-length IgM cDNA along with translated amino acids was shown in Figure S2. The cDNA contains 1980-bp nucleotides, including 40-bp 5′ untranslated sequence and 221-bp 3′ untranslated sequence (with a stop codon, TAG, and a poly-A signal, AATAAA). Alignment of two full-length IgM peptide variants and the reference sequence was shown in Figure 1A. As indicated in the Figure, the constant domain (CH) is fully identical (except for one amino acid substitution, marked with an asterisk, mentioned above), containing CH1–CH4 segments and a C-terminal peptide, encoded by and jointed from separated exons in the germline. The sequences of the variable domain (VH) contain either one or more gaps or substituted amino acids but three complementarity determining regions (CDR1-CDR3) and a joint segment (JH) were recognized. A three-dimensional view of a presentative IgM protein shows that the variable domain takes one arm of the "Y" shape with three CDRs fully exposed (Figure 1B; viewed by Swiss-Model: https://swissmodel.expasy.org/interactive). The two sequences shown in Figure 1A were deposited in GenBank with accession numbers of MZ773892 and MZ773893.

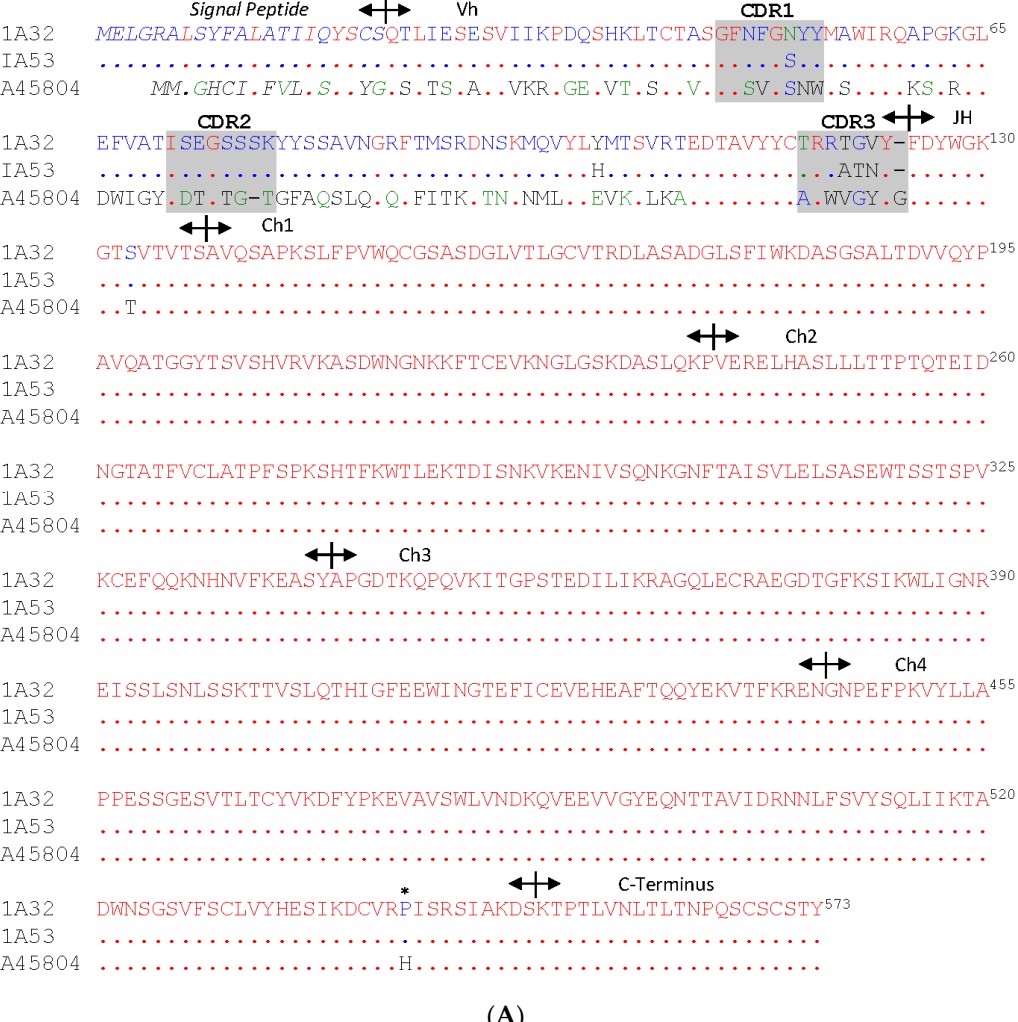

(**A**)

**Figure 1.** *Cont.*

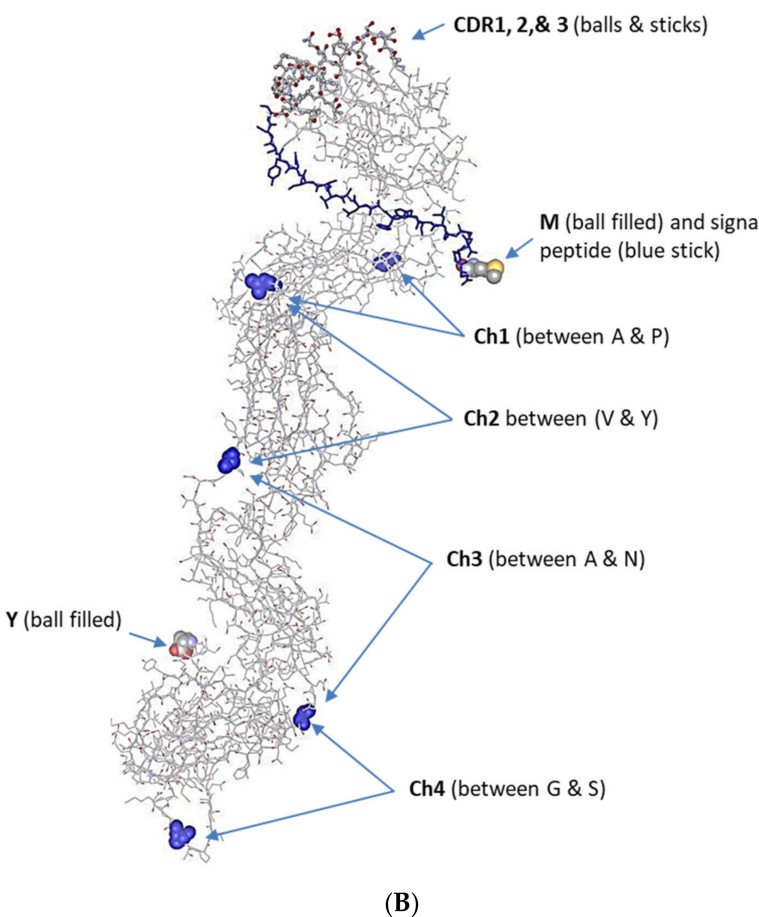

(**B**)

**Figure 1.** (**A**) Alignment of two full-length Ig M heavy chain peptides identified in this study (1A32 and 1A53) with a reference amino acid sequence (accession# A45804). Identical and missing amino acids are indicated by a dot (.) and dash (-), respectively. Signal peptide, variable domain (VH), joint segment (JH), constant domain (CH1, CH2, CH3 and CH4 segments) and C-terminal peptide are shown with arrows delineated. One amino acid substitution in the constant domain is marked with an asterisk (*). Three complementarity determining regions (CDR1–CDR3) are shown in shaded boxes and the diversity (D) segment is within the CDR3. (**B**) The 3-dimension structure view of IgM (1A32) with Swiss-Model. Highlighted are the first and last amino acids (M & Y), signal peptide, complementarity determining regions (CDR1–3) in the variable domain, and constant domain (segments CH1–CH4).

*3.3. Analysis of the Variable Domain of IgM Heavy Chain*

A total of 91 clones that had the IgM variable domain gene were obtained in this study. Among the 91 sequences, 6 of them appeared to be truncated at 5′ end and were excluded in the study (data not shown). Alignment of the 85 full-length cDNA revealed that there were 50 unique sequences (non-identical). Translated amino acid peptides of the 50 sequences showed that all peptides were unique. However, among the 50 variants, there were 5 of them encoded by pseudogenes, which either had a stop codon or were out of the translation frame within CDR3 (data not shown but part of the sequence was shown in Table S1). The BLASTn/p search showed that none of the 45 unique nucleotide/amino acid sequences were identical to sequences deposited in NCBI database. Alignment of the 45 peptides was shown in Figure 2A. The number of amino acids in a variable domain (minus the JH gene segment) varies from 99 to 111. There are only 9 consensus amino acid residues (marked with asterisks) in the variable domain. The complementarity determining regions (CDR) are highly variable and the CDR3 is the most diversified region (Table S1). The overall similarity of these variants is diagramed in Figure 2B. Three frameworks (FR1–FR3) connect with three CDRs with diversity (DH) segment flanked by FR3 and JH. By comparison with

the reference sequences, IGVH families 1–13 [10], the 45 variants apparently fall into 9 of the 13 families (Table 3). In head kidney tissue, there were 16 and 8 unique variants identified in ECP-immunized and mock-immunized fish, respectively; they were grouped to 6 and 4 families, respectively; and most variants in ECP-immunized fish were found in IGVH2. In liver tissue, 15 unique variants were identified in ECP-immunized fish, distributed to 3 families with most (11) in IGVH2 while there were 6 unique variants, distributed to 4 families, identified in the liver tissue of mock-immunized fish. The sequences of variants shown in Figure 2A (except 1A32 and 1A52) were deposited in GenBank with accession numbers of MZ803140-MZ1A803182.

```
                          >   CDR1    <              >   CDR2  <                                            >        CDR3      <    JH
                        *                          *  *                                            * * ****          * ****** * ***
4D9    ---QSLTSSASVVKRPGESVTLSCTVSGFSMG--S--YYMHWIRQKPGKGLEWIGRIDSA-TGTTFAQSLQSQFSITKDTNKNMLYLEVKSLKAEDTAVYYCARRG-GTKY-----FDYWGKGTSVTVTS
4D4    ---VELIQPGSTVLTPGQSMTLTCKVSGYSLTDSS--YCTNWIRQPAGKALEWVGRICSS-GNTYYSEKLKSRFQVSRDTSSSTVTLTGQNMQTEDTAVYYCARQG-VA-------FDYWGKGTTVTVTS
4B7    ---VELTPVTSVMLKPGDSLTLSCKVSGYSVTDNS--YATAWIRQPAGKTLEWINHIWGG-GSNYHKDSLKSKFSISKDGSSSTVTLRGQNLQTEDTAVYYCTRLYSWGSYNA---FDYWGKGTTVTVTS
4B47   --ATELIQPDSVVIKPGETLTITCRVSGASITDRNSHYRTGWIRQPAGKTLEWINSIDYD-GNIYKKDSLKDKFVVSRDTSSDTLTLRGQHMQTEDTVVYYCGRGG-GLGWHYA--FDYWGKGTTVTVTS
4B42   --ATELIQPDSVVIKPGETLTITCRVSGASITDRNSHYRTGWIRQPAGKTLEWINSIDYD-GNIYKKDSLKDKFVVSRDTSSNTLTLRGQNMQTEDTVVYYCGRGG-GLGWHYA--FDYWGKGTTVTVTS
4B41   ---VELIQPGSTLLTPGQSVTLTCKVTGYSLTDSS--YCTHWIRQPAGKALEWVGDICGD-GNTYYSEKLKSRFQVSRDTSSSTVTLTGKNMQTEDTAVYYCAREP-WGVRA----FDYWGKGTTVTVTS
3D9    ---EELTQPSSMTVQPGQSLSINCKVS-YSVT--S--YYTAWIRQPAGKALEWIGNIYSG-GSTAYSDKLKNKFSISRDTATNTITIGGRNLQTEDTAVYYCARLD-IAYY-----FEYWGKGTSVTVTS
3D8    ---VELTQPASMTVQPGQSLSINCKVS-YSVT--S--YDTAWIRQPAGKALEWISYISTG-GSTYYNDKLKNKFSISRDTATNTITIGGQNLQTEDTAVYYCAR----TTYDY---FDYWGKGTSVTVTS
3D7    ---VELIQPGSTLVIPGQSMTLTCKVSGYSSTDNR--YCTHWIRQPAGKALEWIGWICYD-GGTGYSEKLKSRFQISRDTSSSTVTLTGQNMQTEDTAVYYCARQA-ATTD-----FDYWGKGTSVTVTS
3D6    ---EELTQPSSMTVQPGQSLSINCKVS-YSVT--S--YDTAWIRQPAGKALEWIGHIYSG-GSTTYSDKLKNKFSISRDTATNTITIRGQNLQTEDTAVYYCARSG-VALY-----FDYWGKGTSVTVTS
3D5    ---VELIQPGSTVLTPGQSMSLTCKVSGYSLTDSN--YCTAWIRQPAGKALEWIGRICGS-GSTLYSEKLKSSCQVSRDTSSSTVTLTGNNMQTEDTAVYYCAR---ASHYG----FDYWGKGTSVTVTS
3D4    ---VELTQAASMTVQPGQSLSINCKVS-YTVT--S--HDTVWIRQPAGKALEWISLISSGDGSTYYSDKLKNKFSISRDTPTNTITIRGENLQTEDTAVYYCAR----GYYDA---FDYWGKGTTVTVTS
3D3    IKCIALVQPPVMVVKPGESFSVPCKITGYSAT--S--TCTNWIRHKSGQALEWIGWYCSS-SNTGSIDSLKNKLRFSAEASSNTVILHGQNFQSEDTAVYYCAREL-AAGI-----FDYWGKGTSVTVTS
3D14   ---IELTQPASMTVQPGQSLSINCKVS-YSVT--S--DHTAWIRQPAGKALEWIGNIYSS-GDTAYSDKLKNKFSISRDTATNTITIGGQNLQTEDTAVYYCARSP-NGNA-----FDYWGKGTTVTVTS
3D13   ---GELTQPASMTVQPSQSLSINCKVS-YSVT--S--YYTAWIRQPAGKALEWIGYINNN-GGTVYSDKLKNKFTISRDTATNTITIRGQNLQTEDTAVYYCAR-G-PSSYDY---FDYWGKGTSVTVTS
3D12   ---EELSQPASMTVQPGQSLSINCKVS-YSVT--S--YYTVWIRQSAGKALEWIGYISNS-GNTAYSDKLKNKFSISRDTVTNTITIGGQNLHTEDTTVYYCARVR-GGISY----FDYWGKGTSVTVTS
3C8    ---VELIQPGSTVLTPGQSMTLTCKVSGYSLTDSS--YCTGWIRQPAGKALEWIGEICSS-GGTYYSEKLKSRFQVSRDTSSSTVTLTGQNMQTEDTAVYYCARRY-HA------FDYWGKGTTVSVTS
3B46   ---VELTQPASMTVQPGQSLSINCKVS-YSVT--S--YDTAWIRQPAGKALEWISYISSG-GSTYYNDKLKNKFSISRDTATNTITIGGQNLQTEDTAVYYCAR----DYYDY---FDYWGKGTTVTVTS
3B44   ---EELTQPASMTVQPGQSLSINCKVS-YSVT--S--DHTAWIRQPAGKAMQWIGYIHSS-GSTYYNDKLKNKFSISRDTATNTITIGGQNLQTEDTAVYYCARGG-SNA------FDYWGKGTTVTVTS
3B43   ---IELTQPASMTVQPGQSLSINCKVS-YSVT--S--DHTAWIRQPAGKALEWIGNIYSN-GDTAYSDKLKNKFSISRDTATNTITIGGQNLQTEDTAVYYCARSP-NGNA-----FDYWGKGTTVTVTS
3B42   ---VELTQPASMTVQPGQSLSINCKVS-YSVT--N--YHTAWIRQPAGKALEWIGYIHSS-GSTAYSDKLKNKFSISRDTATNTITIGGQNLQTEDTTVYYCARGV-LGALD----FDYWGKGTTVTVTS
2E9    ---EELTQPASMTVQPGQSLSINCKVS-YSVT--S--YHTAWIRQPAGKALEWISLINSG-GSTYYSDKLKNKFSISRDTATNTITIRGQNLQTEDTAVYYCAR---YSGY-----FDYWGKGTQVTVTS
2E8    ---VELIQPGSTVLTPGQSMTLTCKVSGYSLTDSN--YCTHWIRQPAGKALEWVGQICGS-GNTYYSEKLKSRFQVSRDTSSSTVTLTGQNMQTEDTAVYYCAREQ-QLEY-----FDYWGKGTSVTVTS
2E5    ---EELTQPASMTVQPGQSLSINCKVS-YSVT--S--YYTAWIRQPAGKALEWIGYISNS-GGTAYSDKLKNKFSISRDTATNTITIGGQNLQTEDTAVYYCAKIA-AGRWA----FDYWGKGTAVTVTS
2E3    ---QTLIESDSVIIKPDQSHKLTCTASGFNFG--G--SWMAWTRQAPGKGLEWVATITDTSGSKYYSSAVNGRFTMSRDNSKMQVYLHMTSVRTEDTAVYYCTR-R-SGNY-----FDYWGKGTSVTVTS
2E2    ---IELTQPASMTVQPGQSLSINCKVS-YSVT--S--YDTAWIRQPAGKALEWIGNIYSS-GGTAYSDKLKNKFSISRDTATNTITIGGQNLQTEDTAVYYCANTG-VALYNA---FGYWGKGTTVTVTS
2E13   -ATELLQPDSVVIKPGETLTITCRVSGASITDSSSHYGTAWIRQPAGKSLEWINTIYYD-GGINKKDSLKDKFVISRDTSSSTVILTGQNMQTEDTAVYYCARGS-QLRRQDGA-FDYWGKGTTVTVTS
2E1    ---QTLIESDSVIIKPDQSHKLTCTASGLDIS--S--SWMAWIRQAPGKGLEFVAIIESDSDRKFYSNAVNGRFTISRDNSKKQVYLQMNNVRTEDTAVYYCAR-D-PTHYGR---FDYWGKGTSVTVTS
2B12   ---VQLIQPGAMILSPGQSITLTCKVSGYSVSDGS--YWTHWIRQPAGKALEWVGQISGS-GSTYYSEKLKSRFQVSRDTSSSTVTLTGQNMQTEDTAVYYCAREEGGGTY-----FDYWGKGTSVTVTS
1E6    ---VELTQPASMTVQPGQSLSINCKVS-YTVT--S--YDTAWIRQPAGKALEWISLISSG-GSTYYSDKLKNKFSISRDTATNTITIGGQNLQTEDTAVYYCARSG-RVDYNA---FGYWGKGTTVTVTS
1E5    ---EELTQPASMTVQPGQSLSINCKVS-YSVT--S--YYTAWIRQPAGKALEWIGYISNS-GGTTYSDKLKNKFSISRDTATNTITIGGQNLQTEDTAVYYCALRG-WT-------FDYWGKGTAVTVTS
1E4    ---VELTQPASMTVQPGQSLSINCKVS-YSVT--S--YTTAWIRQPAGKALEWIGHINSG-GSTYYSDKLKNKFSISRDTATNTITIGGQNLQTEDTAVYYCAE-S-IGSWR----FDYWGKGTQVTVTS
1E3    ---VELTQPASMTVQPGQSLSINCKVS-YSVT--G--YDTAWIRQPAGKALEWISYISTG-GSTYYNDKLKNKFSISRDTATDTITIGGQNLQTEDTAVYYCAR-E-SYGY-----FDYWGKGTSVTVTS
1E2    ---VELTQPASMTVQPGQSLSINCKVS-YSVT--S--YDTAWIRQPAGKALEWISYISTG-GSTYYNDKLKNKFSISRDTATNTITIGGQNLQTEETAVYYCAR----TSYDY---FDYWGKGTSVTVTS
1E12   --AIDFAQPGHVVVQPGQALTIGCKVSGYSLTDNS--YCTNWIRHATGENMQWIGYICSS-GSTGYTDSLKNRFSITQDTSSNTVTLQGQNVQMEDTAVYYCVRYR-GPDY-----FDYWGKGTSVTVTS
1E11   ---VELIQPGSTVLTPGQSVSLTCKVSGYSLT-ST--YCTHWIRQPAGKTLEWIGWICYD-GSTGYSEKLKSRFQISRDTSSSTVILTGQNMQTEDTAVYYCAR----GAYSA---FDYWGKGTQVTVTS
1E10   ---VELTQTASMTVQPGQSLSINCKVS-YSVT--S--YSTAWIRQPAGKALEWINLIYYD-GDIRQKDSLKNKFVISRDTSSNTVTLQGQNMQTEDTAVYYCAR-F-PSSWSSGYYFDYWGKGTSVTVTS
1E1    ---VEFTQSDNIVVRAGEAFTISCKFSGFSIS--S--YCPCWIRQMPSKTLEYIGYVCGS--SSNVKDSLKSKISFSADVSSSTVFLKGQNFQTEDTAVYYCTRVN-WA-------FDYWGKGTAVTVTS
1C8    ---EELTQPASMTVQPGQSLSINCKVS-YSVT--S--YRTAWIRQPAGKGLEWIGYISSG-GSTAYSDKLKNKFSISRDTATNTITIGGQNLQTEDTAVYYCAGFD-GADLA----FGYWGKGTTVTVTS
1C17   ---QSLTSSASVVKRPGESVTLSCTVSGFSVG--S--NWMSWIRQKPGRGLEWIGYIDTG-TGTGFAQSLQGQFSITKDTNKNMLYLEVKSLKAEDTAVYYCAREL-LGP------FDYWGKGTQVTVTS
1A53   ---QTLIESESVIIKPDQSHKLTCTASGFNFG--S--YYMAWIRQAPGKGLEFVATISEGSSSKYYSSAVNGRFTMSRDNSKMQVYLHMTSVRTEDTAVYYCTR-R-ATNY-----FDYWGKGTSVTVTS
1A32   ---QTLIESESVIIKPDQSHKLTCTASGFNFG--N--YYMAWIRQAPGKGLEFVATISEGSSSKYYSSAVNGRFTMSRDNSKMQVYLMTSVRTEDTAVYYCTR-R-TGVY-----FDYWGKGTSVTVTS
1B35   ---VELTQPASMTVQPGQSLSINCKVS-YTVT--S--YDTVWIRQPAGKTLEWISLIISG-GSTYYSDKLKNKFSISRDTATNTITIRGQDLQTEDTAVYYCARRP-AGRGGA---FDYWGKGTAVTVTS
1B32   ---VQLIQPGTMILSPGQSITLTCKVSGYSVSDGS--YWTHWIRQPAGKALEWVGQISGN-GNTYYSEKLKSRFQVSRDTSSSTVTLTGQNMQTEDTAVYYCARER-DGGDA----FGYWGKGTTVTVTS
1B31   -AEIRLDQSPAVVKRPEETVKISCKINGYDMT--E--HYIHWIRQKPGKALEWVGRMDAGNNNAEYAESLKNQFTLTEDVPASTQYLEAKSLRTEDTAVYYCARRY--GSWSY---FDYWGKGTSVTVTS
```

(**A**)

**Figure 2.** *Cont.*

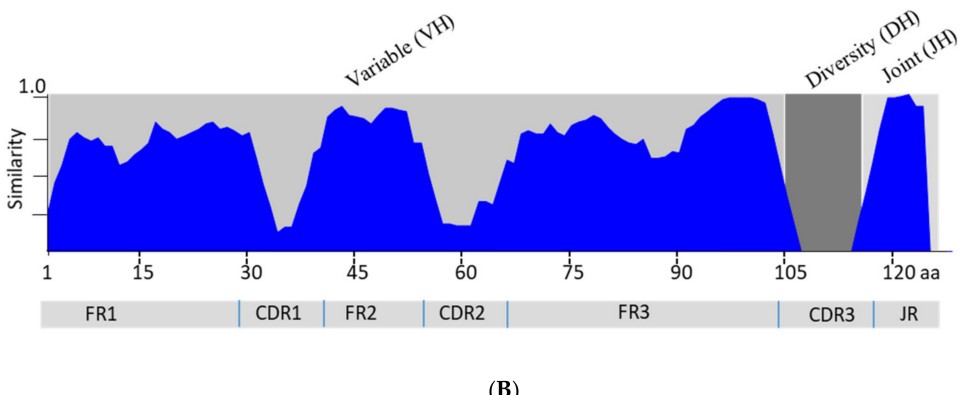

(**B**)

**Figure 2.** (**A**) Alignment of variants of IgM variable domain plus joint segment (JH). Consensus amino acids among all sequences are marked with asterisks. Three complementarity determining regions (CDR1–CDR3) are indicated (between > and <) and the diversity (D) segment is within CDR3. Sequence codes (names) beginning with 1 and 3 are those cloned from ECP-immunized head kidney and liver, respectively; and the codes beginning with 2 and 4 are those cloned from mock-immunized head kidney and live, respectively. (**B**) Diagram of overall similarity of the variable domain (VH) of 45 IGVH variants. Three frameworks (FR1–FR3), three complementarity determining regions (CDR1–CDR3), and a joint (JH) segment are shown. Similarity values range from 0–1 (1 = identical, 0.5 = similar, and 0.2 = weakly similar).

**Table 3.** Variants of IgM heavy chain variable domain cloned from tissues of ECP-immunized and mock-immunized channel catfish and distribution in IGVH families.

| IGVH Family (Reference) [1] | Head Kidney | | Liver | | Variants | |
|---|---|---|---|---|---|---|
| | Immunized | Mock | Immunized | Mock | Unique | Total |
| 1 (AAA79003) | 1 | | | 1 (3) [2] | 2 | 4 |
| 2 (AAA49333) | 9 (14) | 3 (6) | 11 (18) | | 23 | 38 |
| 3 (AAA49334) | 2 (3) | 2 | | | 4 | 5 |
| 4 (AAA49330) | | | | 1 (2) | 1 | 2 |
| 5 (AAA49336) | | 1 (3) | | 2 (5) | 3 | 8 |
| 6 (AAA49328) | 2 | 2 (5) | 3 (4) | 2 (6) | 9 | 17 |
| 7 (EU492631) | 1 (2) | | | | 1 | 2 |
| 9 (AAQ83455) | | | 1 (2) | | 1 | 2 |
| 12 (AAA83460) | 1 (2) | | | | 1 | 2 |
| 13 (pseudo) | [1] | [2] | | [2] | [5] | [5] |
| (Total) | 16 (26) | 8 (18) | 15 (19) | 6 (22) | 45 | 85 |

[1] Additional reference sequences used in this study: AAQ83462 (IGVH8), AAQ83457 (IGVH10), AAQ83459 (IGVH11); IGVH13 was pseudogene (Bengtén et al. [10]). [2] Numbers in parentheses include unique variants and identical sequences of one or more variants; numbers in brackets are pseudogenes (which either had a stop codon or were out of translational frame).

*3.4. Phylogenetic Analysis*

Phylogenetic analysis (Figure 3) revealed that the out-group reference sequences, AAA61292 (from human) and CAA33376 (from horn shark), were apparently more similar than those from channel catfish. The 45 IGVH variants in individual clades basically match IGVH family grouping. Notably, there are 23 variants grouped into a large clade, which contains reference sequence IGVH2, and 20 out of the 23 are variants derived from ECP-immunized head kidney (9) and liver (11) tissues. The IGVH 6 containing clade is the next large group, which has variants from all 4 tissues.

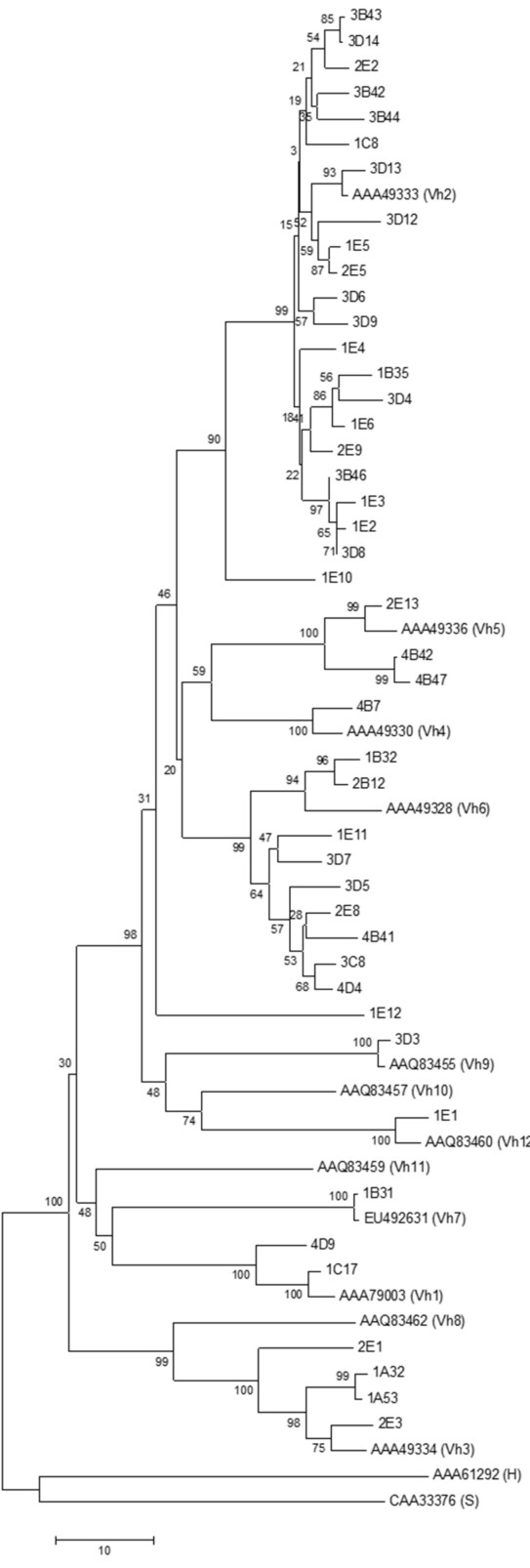

**Figure 3.** Phylogenetic analysis of variants of IgM variable domain with reference sequences (Gen-Bank accession number followed by channel catfish IGVH family based on Bengtén et al. [10]). IgM AAA61292 (H) and horn shark-origin CAA33376 (S) were human- and horn shark-origin, respectively. The tree was constructed using the neighbor-joining method in MEGA6 (numbers shown next to branches are percentages upon a 1000-replicate bootstrap test). The variant ID was illustrated in the legend of Figure 2A.

### 3.5. Two-Dimensional Gel Analysis of IgM Protein

The heavy and light chains of mature IgM protein were well separated on 2-dimensional gel electrophoresis (Figure 4). The apparent molecular weights of heavy and light chains were approximately 66 kDa and 23 kDa, respectively. Based on the calculated molecular weights of 45 variants described above, there is approximately 1.5 kDa difference in heavy chain peptides among the variants. The isoelectric points (pI) of both heavy and light chains were across the range from 6 to 10 while the calculated pI range of the 54 heavy chain variants is between 5.17 and 9.45. By the sizes of dots shown in the gel, some variants appear to be more abundant than others.

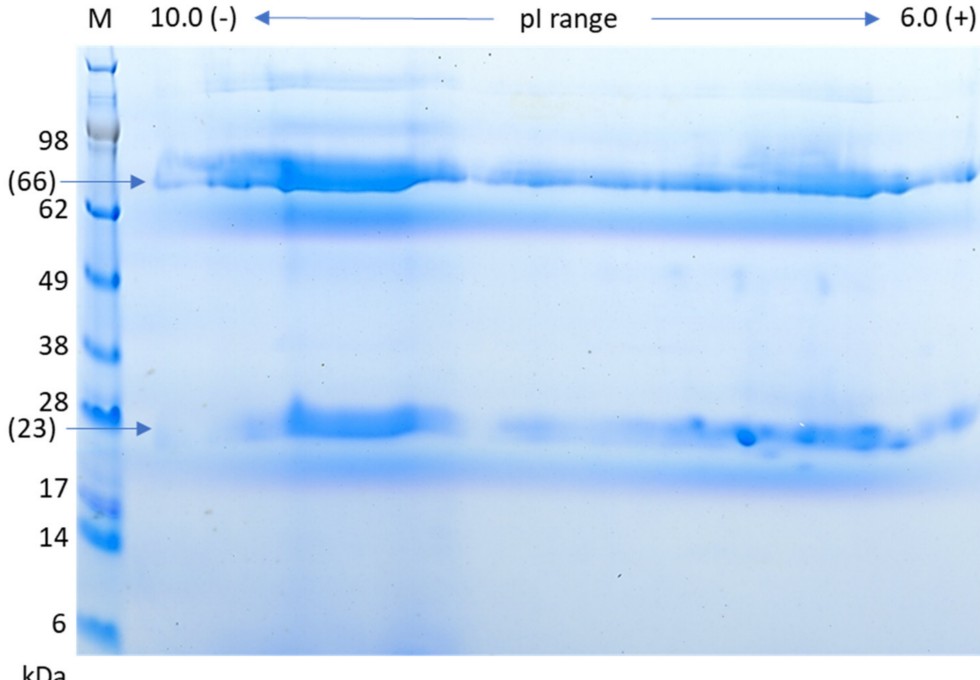

**Figure 4.** Two-dimensional gel electrophoresis of partially purified IgM protein of channel catfish immunized with virulent *Aeromonas hydrophila* ECP antigens. Positions of IgM heavy chain (approximately 66 kDa) and light chain (approximately 23 kDa) were indicated by arrows. M: SeeBlue plus2 pre-stained standards (Invitrogen).

## 4. Discussion

The ability to respond to immunization by the production of specific antibodies is directly related to the repertoire of B cells, which proliferate, produce IgM, and play regulatory functions in teleost systemic immunity [3,18]. By the comparison of IgM variants in tissues of ECP- and mock-immunized fish, results of this study revealed an insight into the differential expression and diversity of IgM in response to immunization.

The result of qPCR analysis of IgM transcripts indicated that there was significantly more IgM produced in fish vaccinated with immunogens. There was approximately a 60% increase in head kidney and a 47% increase in liver. The IgM concentration in teleost fish has been reported to increase more than two-fold in response to environmental stimuli (such as water temperature and quality) and microbes in/on body surface, but specific antibodies increased more when fish were subjected to parasite infection or vaccination [3]. Some specific immunoglobulins may exhibit differential expression patterns among different fish tissues. Perdiguero et al. [19] found that some specific immunoglobulins only expressed in gill and gut but not in spleen of rainbow trout, which was attributed to the direct contact of and response to microbes in the surface of those organs. In analyzing IgM gene expression pattern in European eel (*Anguilla anguilla*), Feng et al. [20] reported that more IgM transcripts were found in kidney than in spleen and other tissues. The earlier report

from Zhang et al. [15] also indicates that the kidney tissue of channel catfish vaccinated with ECP of vAh had significantly more IgM transcripts than the counterpart of mock-immunized fish. The increased IgM transcripts in the kidney and liver observed in this study likely resulted from adaptive immune responses to immunogens.

Variants of the IgM heavy chain V domain characterized in this study were as variable as those previously described in channel catfish [6,10] but vaccinated fish apparently yielded more unique variants than control fish. There were 62% and 79% unique variants in kidney and liver of vaccinated fish, respectively, while there were 44% and 27% variants in kidney and liver of control fish, respectively. Among the unique variants, members in IGVH2 were accounted for approximately 49% and most (approximately 87%) were found in vaccinated fish. This observation was different from what was found by Bengtén et al. [6,10] and Yang et al. [21], in which they observed that most expressed variants were members in IGVH9 and IGVH7, followed by IGVH1, IGVH6, IGVH2, IGVH5, and IGVH10. The difference could be due to the fact that the variants they found were from erythrocytes and spleen tissue of normal (untreated) channel catfish while the variants analyzed in this study were from tissues of immunized fish. Whether members of IGVH2 were immunogen specific variants (or antibodies responsible for binding virulence factors of vAh) has yet to be investigated further since the number of variants identified in this study is still not large enough to draw the conclusion. Nevertheless, this is the first study, to our knowledge, in characterization of expressed IgM variants of channel catfish in response to vAh immunization and revealing specific, antigen-driven variants.

Structural diversity of IgM variants is determined by the somatic recombination of variable (*V*), diversity (*D*), and joint (*J*) gene segments [13,22,23]. The unique variants in a heavy chain V domain identified in this study were highly variable not only between members of IGVH families but also among members of the same family. While variation between different families was derived from germline *V* gene segments, the variation of members in a family could be due to the multiple segments of the same family gene in a genome, resulting from gene-containing block duplication, transposition and mutation [10]. The least sequence similarity among all variants was in the third complementarity determining region (CDR3) segments, where the *D*-gene segment was joined by random non-templated nucleotides with *V*-gene and *J*-gene segments (Supplemental Table S1) [24]. This process, named somatic hypermutation and antigen selection, defines the specificity of an antibody; detailed analysis of the recombination process would facilitate further understanding of how antibodies develop against antigens of interest [12,13].

As reported in earlier studies [6,25], some expressed IgM transcripts analyzed in this study were identified as pseudogenes due to translation frame shifts or the presence of in-frame stop codon. The process of expression of pseudogenes may be due to the random nature of the gene recombination event; or it could be resulted from a gene conversion mechanism [6]. Whether the allelic counterpart of a pseudogene is functional in other individuals [6] has yet to be investigated.

IgM protein analysis in 2D gel revealed that IgM protein variants were as variegated as cDNA variants. Amino acid variation in V domain peptides greatly affect the isoelectric points of individual IgM variants; the observed pI values in protein gel were similar to the calculated pI values from translated IgM transcripts. The apparent molecular weight (MW) of heavy chain revealed in protein gel was higher than that of the calculated MW; this discrepancy is likely due to the heavy glycosylation of the heavy chain of mature antibodies as observed in other teleost fish [26]. It would be of interest to analyze differential 2D profiles of IgM in fish immunized with individual specific antigens to identify unique antibodies.

## 5. Conclusions

In conclusion, channel catfish developed adaptive immunity against virulent *Aeromonas hydrophila* upon vaccination with extracellular proteins of the pathogen. The antibody-mediated immune response was revealed by production of specific immunogen-driven IgM variants. Results of this study provided insight into the molecular and genetic

basis of antibody diversity, enriched our knowledge of the complex interplay between antigens and antibodies and would facilitate future research for decoding the mechanism of tailoring antigen receptor specificity.

**Supplementary Materials:** The following are available online at https://www.mdpi.com/article/10.3 390/fishes7010024/s1, Figure S1: Analysis of qPCR amplicons. Figure S2: The full-length cDNA sequence and translated amino acid of catfish IgM heavy chain (variant 1A32). Table S1: The third complementarity region (CDR3) of IgM heavy chain variable domain (IGVH) identified in this study.

**Author Contributions:** Conceptualization, D.Z., M.D.L., C.A.S. and B.H.B.; methodology and investigation, D.Z.; data analysis, D.Z. and M.D.L.; writing—draft preparation, D.Z.; writing—review and editing, M.D.L., C.A.S. and B.H.B. All authors have read and agreed to the published version of the manuscript.

**Funding:** This research was supported by the U.S. Department of Agriculture, Agricultural Research Service (CRIS project #6010-32000-027-00D).

**Institutional Review Board Statement:** The study involving fish was approved by the Institutional Animal Care and Use Committee of the USDA-ARS Aquatic Animal Health Research Unit.

**Data Availability Statement:** The data that support the findings of this study will be available on request to the corresponding author.

**Acknowledgments:** We thank D. Xu and N. Qin for their technical support of some experiments in this study.

**Conflicts of Interest:** The authors declare no conflict of interest with any financial/research/academic organization. The USDA is an equal opportunity provider and employer. Mention of trade names or commercial products in this publication is solely for the purpose of providing specific information and does not imply recommendation or endorsement by the United States Department of Agriculture.

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
