# Peer review of "Identification and Characterization of Differentially Expressed IgM Transcripts of Channel Catfish Vaccinated with Antigens of Virulent Aeromonas hydrophila"

_fishes, doi:10.3390/fishes7010024_

Round 1

Reviewer 1 Report

The manuscript with ID (fishes-1522153) by Zhang and coauthors identified and characterized differentially expressed IgM transcripts from Channel catfish vaccinated with Antigens of Virulent Aeromonas hydrophila. The study is interesting; however, I find Minor Revisions to be carefully amended by authors before the manuscript is considered for publication in Fishes.

Line 11: Aeromonas hydrophila was isolated from diseased fish many years ago. Remove “in recent years”

Line 15: indicated that

Line 16: significantly increased

Line 38: Three isotypes of Ig

Line 39: write which species of catfish?

Line 429: Aeromonas hydrophila (italic)

Line 437: Gadus morhua (italic)

Author Response

Reviewer #1
Line 11: Aeromonas hydrophila was isolated from diseased fish many years ago. Remove “in recent years”
- Revised: Line 11.
Line 15: indicated that
- Revised: Line 15.
Line 16: significantly increased
- Comments: We think that it’s appropriate to state “produced significantly more”.
Line 38: Three isotypes of Ig
- Revised: Line 39.
Line 39: write which species of catfish?
- Revised: Line 41.
Line 429: Aeromonas hydrophila (italic)
- Corrected: Line 434.
Line 437: Gadus morhua (italic)
- Corrected: Line 442.

Reviewer 2 Report

General comments:

This manuscript analyzed IgM transcripts that were differentially expressed in the kidney and liver of vAh extracellular proteins (ECP)-immunized and mock-immunized (control) channel catfish with emphasis on variable domain (VH) of heavy chain. qPCR indicated immunized fish produced significantly more IgM transcripts than control fish. Full-length IgM heavy chain cDNA was cloned and great sequence diversity was revealed in VH segment of IgM peptide. Two-dimensional gel electrophoresis of semi-purified IgM protein confirmed that matured IgM protein was as variable as IgM transcripts identified. The results in this study would provide basic information to understand the molecular and genetic basis of antibody diversity in Ictalurid catfish.

The experiment design of the manuscript is reasonable and English expression is relatively fluent, but the data analysis are not reasonable (see below). In addition, some statements in the M&Ms are incomplete. Therefore this manuscript need a minor revision at present before acceptance for publication.

Other comments:

1. Introduction

Line 35: please use “teleosts” in place of “teleost fish”.

2. M&Ms

Line 75-78: how many individuals were challenged? What was the dose of vAh? What was the detailed tissue collecting method, such as the number of replicates in each experimental group, the number of fish individuals in each replicate? 

Line 87: “Quantitative PCR” should be “Quantitative real-time PCR”.

Line 111: please give the full name of “LB” when it first appears.

Line 118: “Quantitative PCR” should be “Quantitative real-time PCR”.

Line 122: Why did the authors use 18S rRNA gene as an internal gene? Did you compare it with other genes?

Line 166: Did you use one-way ANOVA to compare the difference among means in this study? I think that this is inconsistent with the analysis used in Table 2 in the results.

3. Results

Line 177: In table 2, the comparison of means between ECP- and mock-immunized groups should be performed using t-test instead of ANOVA. Please correct the error of the data analysis method in the section 2.6.

4. Discussion

Line 367: What further studies will be needed to perform in the future?

Author Response

Reviewer #2:
1.Introduction
Line 35: please use “teleosts” in place of “teleost fish”.
- Revised: Line 36.
2.M&Ms
Line 75-78: how many individuals were challenged? What was the dose of vAh? What was the detailed tissue collecting method, such as the number of replicates in each experimental group, the number of fish individuals in each replicate?
- Revised: Lines 76-79.
Line 87: “Quantitative PCR” should be “Quantitative real-time PCR”.
- Revised: Line 89.
Line 111: please give the full name of “LB” when it first appears.
- Revised: Line 114.
Line 118: “Quantitative PCR” should be “Quantitative real-time PCR”.
- Revised: Line 120.
Line 122: Why did the authors use 18S rRNA gene as an internal gene? Did you compare it with other genes?
- Comments: 18S rRNA is commonly used as a reference gene for qPCR based detection and quantification since it expresses highly and stably under in an organ under the same conditions. It may not be suitable as a reference gene only if the gene of interest (target gene) has very low level expression. It has been assessed in many publications, such as Kosakyan et al. (2019) Sci. Rep.9, 15073. In our previously reported (Ref. #15), twe found that he relatively expression of 18S RNA and IgM was comparable.
Line 166: Did you use one-way ANOVA to compare the difference among means in this study? I think that this is inconsistent with the analysis used in Table 2 in the results.
- Corrected: it’s paired t test (Line 168).
3.Results
Line 177: In table 2, the comparison of means between ECP- and mock-immunized groups should be performed using t-test instead of ANOVA. Please correct the error of the data analysis method in the section 2.6.
- Corrected (see above).
4. Discussion
Line 367: What further studies will be needed to perform in the future?
- Revised: Lines 370-372.

Reviewer 3 Report

In this paper, Zhang et.al analyzed different IgM transcripts that expressed in the kidney and liver of ECP-immunized and mock-immunized (control) channel catfish with emphasis on variable domain of heavy chain. Full length IgM heavy chain cDNA was cloned. Interestingly, vaccinated fish apparently had more unique variants than in control fish. Approximately 49% unique variants were found in VH2 family. Their results here provided insight into the molecular and genetic basis of antibody diversity and enriched our knowledge of the complex interplay between antigens and antibodies in channel catfish.

Major comments

  1. The spleen and head kidneys are also vital tissues for antibody immunity. It would be better if relevant data could be supplemented?
  2. Whether there is a difference in the analysis of the IgM heavy chain variable domain sequence from the liver or kidney?

Minor comments

  1. Line73: How many experimental fish in this study?
  2. Line75: How much is the specific of ‘fatal dose’?
  3. Line77, Line143: How many replicate groups?
  4. Line172 of ‘kidney and liver’ is different from ‘Head kidney and liver’ in Table 2. So what tissues were collected in this study?
  5. Line105-107, Line126-127, Line135: ‘m’ is usually written as ‘min’. Authors should pay attention to the correct format.
  6. Discussion about diversity of IgM variants are supposed to be more abundant.

Author Response

Reviewer #3
Major comments
The spleen and head kidneys are also vital tissues for antibody immunity. It would be better if relevant data could be supplemented?
- Comments: We reported the IgM variations in tissues of liver and head kidney. The spleen is reportedly to be an important organ for antibody immunity; however, no data is currently available for that.
Whether there is a difference in the analysis of the IgM heavy chain variable domain sequence from the liver or kidney?
- Comments: We didn’t see variation of IgM heavy chain between the two tissues from transcripts we sequenced. This observation is in accordance with other published studies.

Minor comments
Line73: How many experimental fish in this study?
- Revised: There were two replicates with 12 fish each (Lines 78-79).
Line75: How much is the specific of ‘fatal dose’?
- Revised: The dosage used was approximately 2 x 107 colony forming units per fish (Lines 76-77).
Line77, Line143: How many replicate groups?
- Revised: Lines 78-79.
Line172 of ‘kidney and liver’ is different from ‘Head kidney and liver’ in Table 2. So what tissues were collected in this study?
- Yes, the anterior (head) kidney is different from posterior (trunk) kidney. In this study, only head kidney was used; we have revised the text using the full term “head kidney”.
Line105-107, Line126-127, Line135: ‘m’ is usually written as ‘min’. Authors should pay attention to the correct format.
- Corrected.
Discussion about diversity of IgM variants are supposed to be more abundant.
- Comments: In this study, we can only assess the diversity of IgM from vaccinated and un-vaccinated fish since there is not enough data of similar study in literature (including gene sequences in data bases) for further comparison. It is expected that there will be more diversified antibody production in response to different antigens.  This study just sheds a light onto this kind of investigation.